# Imaging Phenotypes and Evolution of Hepatic Langerhans Cell Histiocytosis on CT/MRI: A Retrospective Study of Clinical Cases and Literature Review

**DOI:** 10.3390/bioengineering10050598

**Published:** 2023-05-16

**Authors:** Luwen Hao, Yuanqiu Li, Ziman Xiong, Yuchen Jiang, Xuemei Hu, Daoyu Hu, Zhen Li, Yaqi Shen

**Affiliations:** Department of Radiology, Tongji Hospital, Tongji Medical College, Huazhong University of Science and Technology, Wuhan 430032, China; luwenhao@hust.edu.cn (L.H.);

**Keywords:** Langerhans cell histiocytosis, liver involvement, evolution, magnetic resonance imaging, imaging phenotypes, evaluation

## Abstract

(1) Background: pathological changes in hepatic Langerhans cell histiocytosis (LCH) have been observed; however, corresponding imaging findings can appear vague to physicians and radiologists. The present study aimed to comprehensively illustrate the imaging findings of hepatic LCH and to investigate the evolution of LCH-associated lesions. (2) Methods: LCH patients with liver involvement treated at our institution were retrospectively reviewed along with prior studies in PubMed. Initial and follow-up computed tomography (CT) and magnetic resonance imaging (MRI) were systematically reviewed, and three imaging phenotypes were created based on the lesion distribution pattern. Clinical features and prognoses were compared among the three phenotypes. Liver fibrosis was evaluated visually on T2-weighted imaging (T2WI) and diffusion-weighted imaging (DWI), and apparent diffusion coefficient (ADC) values of the fibrotic areas were measured. Descriptive statistics and a comparative analysis were used to analyze the data. (3) Results: based on the lesion distribution pattern on CT/MRI scans, patients with liver involvement were categorized as the disseminated lesion phenotype, scattered lesion phenotype, and central periportal lesion phenotype. Patients with scattered lesion phenotype were typically adults, and only a few of them had hepatomegaly (n_present_ = 1, 1/6, 16.7%) and liver biochemical abnormalities (n_present_ = 2, 2/6, 33.3%); patients with central periportal lesion phenotype were typically young children, and hepatomegaly and biochemical abnormalities were more apparent in these patients than those with another phenotype; and those with the disseminated lesion phenotype were found in all age groups, and the lesions evolved rapidly on medical imaging. Follow-up MRI scans show more details and can better document the evolution of lesions than CT. T2-hypointense fibrotic changes, including the periportal halo sign (n_present_ = 2, 2/9, 22.2%), patchy liver parenchyma changes (n_present_ = 6, 6/9, 66.7%), and giant hepatic nodules close to the central portal vein (n_present_ = 1, 1/9, 11.1%), were found, while fibrotic changes were not observed in patients with the scattered lesion phenotype. The mean ADC value for the area of liver fibrosis in each patient was lower than the optimal cutoff for significant fibrosis (METAVIR Fibrosis Stage ≥ 2) in a previous study that assessed liver fibrosis in chronic viral hepatitis. (4) Conclusions: The infiltrative lesions and liver fibrosis of hepatic LCH can be well characterized on MRI scans with DWI. The evolution of these lesions was well demonstrated on follow-up MRI scans.

## 1. Introduction

Langerhans cell histiocytosis (LCH) is a rare inflammatory and myeloproliferative neoplasm characterized by the proliferation and accumulation of histiocytes that express CD1a, Langerin (CD207^+^), and S-100 proteins [1,2]. These histiocytes may infiltrate any bodily organ or system. In children (≤15 years) and adult (>15 years) patients, the annual incidence is approximately 5–9/10^6^ and 1/10^6^, respectively [3]. The natural progression of liver involvement is divided into two stages, early infiltration of histiocytes and late fibrosis [4], and early liver fibrosis may be reversible [5]. As one of the at-risk organs, liver involvement is related to a poor prognosis [6,7] since hepatic LCH may result in secondary sclerosing cholangitis (SC) and even liver function failure. Transplantation may be needed at late stage [2].

The early diagnosis of LCH can aid in appropriate therapy [8]. Computed tomography (CT) and magnetic resonance imaging (MRI) findings correlate with various histological stages of LCH lesions [9,10]. At present, the main sources of evidence regarding liver involvement are case reports and small case series limited to adults [4,11,12,13,14,15,16,17,18,19,20,21,22,23] or children [9,24,25,26,27,28,29]. Periportal abnormalities and parenchymal lesions have been reported separately in most cases, and there is a lack of comprehensive summary of imaging findings. Previous studies described the fibrotic stage of liver involvement as intrahepatic bile duct stenosis and dilatation on magnetic resonance cholangiopancreatography (MRCP), in addition to the nodule-like appearance seen in advanced liver cirrhosis [10,27]. In a few case reports, T2-hypointense fibrosis in the liver parenchyma has been reported [11,28]. To date, a comprehensive interpretation of MRI findings of histiocyte infiltration and liver fibrosis has not been published.

A liver biopsy is considered the gold standard for assessing liver fibrosis [30]. As an invasive exam, it has several limitations, including sample variability, observer variability, and possible complications, and is, therefore, not an ideal tool for screening, long-term follow-up, or the evaluation of treatment responses [31,32]. Diffusion-weighted imaging (DWI) is widely used to detect and characterize focal and diffuse liver lesions [33,34]. Previous studies have revealed the feasibility of using DWI in the fibrosis staging and follow-up of patients with liver fibrosis [35,36]. In this regard, DWI is utilized to evaluate primary cholangitis fibrosis [37] and chronic viral hepatitis liver fibrosis [38]. As of yet, DWI has not been utilized in the evaluation of hepatic LCH.

The present study retrospectively analyzed the clinical and imaging data of patients treated at Tongji Hospital of Huazhong University of Science and Technology, and compared them with prior studies, which was aimed to document different image phenotypes and the evolution of hepatic LCH, and preliminarily evaluated late fibrosis using DWI.

## 2. Materials and Methods

### 2.1. Patients

The local institutional review board approved this study (No. TJ-IRB20221208) and informed consent was waived for the study’s retrospective nature. A thorough and systematic review of medical records of patients at our institution with pathologically confirmed LCH was performed for cases between January 2014 and July 2022. The inclusion criteria were as follows: (1) the diagnosis of LCH was confirmed by surgery or biopsy pathology (positive CD1a and/or CD207 (Langerin) histiocytes) [1,7]; (2) liver involvement was defined as one or more of the following (based on the guidelines derived from the Euro-Histio-Net project 2012 edition for pediatric patients and 2013 edition for adult patients): abnormal liver biochemistry (total protein, albumin, bilirubin, alanine transaminase aminotransferase (ALT), aspartate transaminase (AST), alkaline phosphatase (AP), γ-glutamyl transferase (γGT)); unexplained liver enlargement; or abnormal intrahepatic lesions revealed by ultrasound (US), CT, MRI, and positron emission tomography-CT (PET-CT)) [6,7]. The exclusion criteria were as follows: (1) patients without relevant liver CT or MR scans; (2) patients with normal abdominal CT or MRI (images were assessed jointly by two senior abdominal radiologists); and (3) critical clinical data used to evaluate liver involvement were unavailable.

Additionally, we conducted a thorough and systematic review of PubMed for studies published in English between January 1992 and October 2022. The search was performed using the primary search terms “hepatic Langerhans cell histiocytosis” and “Langerhans cell histiocytosis, liver involvement”. Cases of liver involvement with descriptions of CT/MRI findings were included in the reference group. A comparative analysis of the clinical features between the present study and the reference group was performed. Cases with lesions illustrated by relevant CT/MRI images in the literatures were analyzed as well. As such, the numbers of patients from present group, reference group, and the combined numbers were recorded as n_present_, n_reference_, and n_combined_.

### 2.2. CT and MRI Examinations

Patients at our institution underwent routine medical imaging scans of the abdomen, including non-contrast CT, dynamic contrast-enhanced CT (CECT), MRCP, unenhanced MRI (routine T1-, and axial and coronal T2-weighted images), DWI with corresponding apparent diffusion coefficient (ADC) maps, and dynamic contrast-enhanced (DCE) MRI. Examinations were performed using three CT (LightSpeed16, GE, USA; Aquilion ONE, TOSHIBA, Tokyo, Japan; Aquilion PRIME, TOSHIBA, Tokyo, Japan) and four MRI scanners (1.5 T Signa Excite, GE, USA; 1.5 T Optima MR360, GE, USA; 3.0 T Discovery MR750, GE, USA; 3 T MAGNETOM Skyra, Siemens, Munich, Germany).

DCE-MRI was not utilized with children; the optimal CT protocol utilized with children was based on age and weight. Radiation doses were below the weight-specific diagnostic reference levels (DRLs) recommended in Radiation Protection No. 185 [39].

### 2.3. Image Analysis

The CT and MRI data from the first through last follow-up were reviewed by two abdominal radiologists (with more than 10 years of clinical CT and MRI experience). They were aware of the purpose of the present study but were blinded to the clinical data. The two radiologists separately analyzed the images from the present study group, and the images illustrated in figures from the reference group. If there were any inconsistencies in the analysis, a consensus was reached after consultation with each other. The following were the primary analytical contents considered: (1) the range of liver lesions, including periportal abnormalities and parenchymal lesions [26]; (2) density/signal intensity, margin, morphology, enhancement pattern, and enhancement degree of the liver lesions; (3) abnormalities of the intrahepatic bile duct; (4) evolution of liver lesions during follow-up; and (5) evidence of fibrosis, which included conspicuous hypointense abnormalities on non-enhanced images that appeared patchy, nodular, or confluent, distributed diffusely in the liver or centered around the portal tracts [11,28,40]. The ADC measurements were obtained from the ADC maps from the final follow-up with b-values of 1000 s/mm^2^. Following Charatcharoenwitthaya et al. [38], three circular regions of interest (ROIs) (20 ± 5 mm^2^) were manually positioned on the hypointense area by each radiologist, respectively. ROIs were also placed on areas based on color hue corresponding to the reduced ADCs in patients without fibrotic changes, which served as controls (all ROIs should be placed on the right lobe to avoid early infiltration lesions, central major vessels and artifacts [41]). The intra-reader agreement was assessed with the original measurement, and ADC values of six ROIs were averaged for the final ADC value for each subject.

### 2.4. Statistical Analysis

Descriptive statistics were used to analyze the clinical and imaging data. Available or missing data from the patients in the present study group were checked, and the available data from all of the reference patients were also included in the analysis. The results are presented as mean ± standard deviation (SD), range, and percentage. Pearson’s chi-square test, Fisher’s exact test, or independent samples *t*-test were used as appropriate to compare the basic characteristics and imaging signs. The intraclass correlation coefficient (ICC) was used to assess the intra-reader agreement of the ADC measurements. All statistical analyses were performed using SPSS (version 25.0).

## 3. Results

### 3.1. Patients and Clinical Characteristics

Twenty LCH patients with liver involvement were enrolled in the present study (Figure 1). Among them, 18 patients underwent CT and 15 underwent MRI. Patients (n_present_ = 12) with follow-up MRI scans were longitudinally reviewed to assess lesion evolution. Patients underwent CT/MRI because of suspected liver involvement from US or PET/CT imaging, or liver biochemical abnormalities. Detailed imaging examinations are listed in Appendix A.

For comparison with the present study group, 24 articles with 67 patients with comparatively eligible data were enrolled in the reference group [4,9,11,12,13,14,15,16,17,18,19,20,21,22,23,24,25,26,27,28,29,42,43,44] (Figure 1). Due to missing data in the reference group, available data on hepatomegaly (n_reference_ = 47, n_combined_ = 67), biochemical liver abnormalities (n_reference_ = 53, n_combined_ = 73), and stratification (n_reference_ = 61, n_combined_ = 81) were analyzed separately. In previous studies, patients underwent CT, MRI, US, and PET/CT. The imaging modalities and sequences obtained among patients were not uniform (Appendix A). Among the 28 patients whose CT/MRI scans were illustrated in prior articles, 24 had infiltrative lesions, and 5 had fibrotic signs demonstrated with images (Appendix A).

The clinical features of the two groups are listed in Table 1. Overall, no significant differences in clinical characteristics were observed between the two groups. Combining the two groups of data, isolated liver involvement was found to be rare (n_combined_ = 3/81, 3.7%), and the majority of patients had multisystem LCH (n_combined_ = 78/81, 96.3%); there were more adult patients (n_combined_ = 52/87, 59.8%) with liver involvement than pediatric patients (n_combined_ = 35/87, 40.2%). The male-to-female ratio was 1.7:1. Not all patients with abnormalities detected on CT or MRI scans presented with obvious clinical manifestations; hepatomegaly and liver biochemical abnormalities were absent in 25 (n_combined_ = 25/67, 37.3%) and 19 (n_combined_ = 19/73, 26%) patients, respectively.

### 3.2. Image Phenotypes Based on Lesion Distribution

The following three image phenotypes were created based on lesion distribution and morphology (Figure 2): the disseminated lesion phenotype (n_present_ = 8, 8/20, 40%); scattered lesion phenotype (n_present_ = 6, 6/20, 30%); and central periportal lesion phenotype (n_present_ = 6, 6/20, 30%). The disseminated lesion phenotype (Figure 3 and Appendix A) is characterized by multiple irregular, patchy, and roundish (size varies, small lesions can be miliary) lesions diffusely distributed throughout the liver parenchyma and portal tracts; the scattered lesion phenotype (Appendix A) is characterized by several (<15) focal lesions of the liver parenchyma randomly scattered without periportal abnormality; and the central periportal lesion phenotype (Figure 4) is characterized by irregular periportal space widening (parenchymal lesions were not found).

Liver biopsy was performed in seven patients, pathological images of three patients (classified into the three image phenotypes, respectively) were available (Appendix A). Based on morphological changes and lesion distribution, the pathological findings of the three patients were consistent with the image phenotype.

The basic characteristics of patients in the present study, along with the three phenotypes, are summarized in Table 1. Additionally, a comparison of the major liver biochemical indicators among the phenotypes is listed in Table 2. There was a statistically significant difference in the distribution of age of onset among the three phenotypes of patients (Appendix A). The patient age of the disseminated lesion phenotype had a wider zone (19.7 ± 12.9 Y, range 1.3–39 Y). The central periportal lesion phenotype was observed in children (1.8 ± 0.4 Y). The scattered lesion phenotype was found in adults (36.8 ± 8.4 Y). Hepatomegaly (n_present_ = 1, 1/6, 16.7%) and liver biochemical abnormalities (n_present_ = 2, 2/6, 33.3%) were found in only a few patients with scattered lesion phenotype. Statistical differences were found in the biochemical indicators (γGT/AP, ALT/AST, and bilirubin) among the three phenotypes in the present study group (Table 2). Biochemical abnormalities in the central periportal lesion phenotype were more frequent than in the other two phenotypes, even though significant differences in biochemical indicators (γGT/AP, *p* = 0.002; ALT/AST, *p* = 0.009; bilirubin, *p* = 0.015) were only found between the central periportal lesion phenotype and scattered lesion phenotypes after multiple comparisons. Most patients had multisystem LCH, while only one (n_present_ = 1, 1/20, 5%) had isolated liver involvement, and presented as the disseminated lesion phenotype on images.

In the reference group, clinical cases (n_reference_ = 24) with lesions demonstrated by imaging figures were analyzed and classified into the three phenotypes. Compared with the present study group, no significant differences of patient age were observed (Appendix A). Combining the data of both groups (n_combined_ = 44), the disseminated lesion phenotype was found in all ages, while the central periportal lesion phenotype was typically seen in young children, and the scattered lesion phenotype was typically found in adults; two patients (n_combined_ = 2, 2/44, 4.5%) were single-system LCH (SS-LCH) and both presented as a disseminated lesion type. 

### 3.3. Detailed Imaging Findings and Evolution of Lesions

#### 3.3.1. Parenchymal Lesions

On CT/MRI scans of 20 patients in the present study group, parenchymal lesions were observed in 14 (14/20, 70%), including the scattered lesion phenotype (n_present_ = 6, 6/20, 30%) and the disseminated lesion phenotype (n_present_ = 8, 8/20, 40%). No parenchymal lesions were found in the central periportal lesion phenotype. Irregular or roundish parenchymal lesions included patchy and miliary (n_present_ = 10, 10/20, 50%) as well as cyst-like (n_present_ = 14, 14/20, 70%) abnormal signal/density lesions of varying sizes (Appendix A). Patchy and miliary lesions with blurred edges showed slightly low signal intensity on T1WI, slightly high signal intensity on T2WI, low density on CT, slightly high signal intensity on DWI, and mild to moderate enhancement. Cyst-like lesions can be divided into thick- and thin-walled on MRI scans, and their detailed characteristics are shown in Figure 2 and Appendix A. CT is limited in the detection and characterization of small lesions and subtle changes (Figure 3). 

Evolution of parenchymal lesions in eight patients with multiple MRI scans

MRI scans were analyzed to reveal lesion evolution as an advantage for the better characterization of small lesions and subtle changes. The imaging findings of the initial and final MRI scans are listed in Appendix A. On initial MRI scans, patchy and miliary lesions were identified in six patients (6/8, 75%), and cyst-like lesions were identified in six patients (6/8, 75%), including thick- (5/8, 62.5%) and thin-walled cyst-like lesions (3/8, 37.5%). 

On the final MRI scans, patchy and miliary lesions were identified in one patient (1/8, 12.5%), and cyst-like lesions were identified in six patients (6/8, 75%), including thick- (2/8, 25%) and thin-walled cyst-like lesions (6/8, 75%). Patchy and miliary lesions regressed until they disappeared, and thick-walled cyst-like lesions disappeared or transformed into thin-walled cysts. Overall, the lesions of the eight patients showed regression or complete resolution. 

Intermediate multiple follow-up scans illustrated subtle changes: thick-walled cysts that developed from patchy and miliary lesions were observed in six patients (6/8, 75%); enlarged and increased patchy and miliary lesions with blurred edges can be observed in a patient whose condition temporarily progressed during initial systemic chemotherapy (the patient’s condition improved after the chemotherapeutic agents were modified).

#### 3.3.2. Periportal Abnormalities

On the CT/MRI scans of 20 patients in the present study group, periportal abnormalities were observed in 14 patients (Appendix A). The central periportal lesion phenotype (n_present_ = 6, 6/20, 30%) with band-like periportal space widening presented as slightly hypointense on T1WI, hypo- to slightly hyperintense on T2WI, slightly low density on CT, and mild-to-moderate enhancement on enhanced sequences. Disseminated lesions (n_present_ = 8, 8/20, 40%) had slightly irregular periportal space widening with periportal patchy or miliary lesions. Periportal abnormalities were not observed in the scattered lesion phenotype. MRI provided better details of lesions and follow-up changes than CT.

Evolution of periportal lesions in eight patients with multiple MRI scans

For the disseminated lesion phenotype (n_present_ = 4, 4/8, 50%), the evolution of periportal patchy or miliary lesions was the same as that of liver parenchymal lesions.

For the central periportal lesion phenotype (n_present_ = 4, 4/8, 50%), initial scans revealed slight periportal hyperintensity in two patients (2/8, 25%) or mixed hypo- and hyperintensity in two patients (2/8, 25%) on T2WI/DWI. On follow-up MRI scans, the regression of lesions, in which the thickness of periportal abnormality contracted and signal intensity decreased on T2WI/DWI, was revealed in two patients (2/8, 25%) (Figure 4). The lesions remained stable in the other two patients. 

#### 3.3.3. Hepatic Fibrosis and Other Imaging Findings

Reviewing all imaging data of each patient in the present study group, some of the accompanying imaging findings are shown in Table 3.

Among the 15 patients who underwent MRI examination, hypointense changes on T2WI/DWI were observed in 9 patients (9/15, 60%). The manifestations appeared in three patterns (Figure 2 and Appendix A): periportal halo sign (n_present_ = 2, 2/9, 22.2%), patchy liver parenchyma changes (n_present_ = 6, 6/9, 66.7%), and giant hepatic nodules close to the central portal vein (n_present_ = 1, 1/9, 11.1%). Among the nine patients with T2WI/DWI hypo-signal changes on behalf of hepatic fibrosis, six (6/9, 66.7%) developed from the central periportal lesion phenotype and the other three (3/9, 33.3%) developed from the disseminated lesion phenotype; however, T2WI/DWI hypo-signal changes were not observed in the scattered lesion phenotype. All nine patients presented with compression deformation of the central portal tracts and sparse peripheral portal tracts.

The mean ROI-ADC value of each patient with the hypointense changes on T2WI/DWI was less than 1.04 × 10^−3^ s/mm^2^, the optimal cutoff for significant fibrosis (METAVIR Fibrosis Stage ≥ 2) reported by a previous study using DWI to assess liver fibrosis in chronic viral hepatitis [38]. The mean ROI-ADC value per patient without this sign as a control was greater than 1.04 × 10^−3^ s/mm^2^ (ICC = 0.60) (Appendix A).

Among the 20 patients in the present group, 13 (13/20, 65%) had hepatomegaly, 12 (12/20, 60%) had splenomegaly, and 11 (11/20, 55%) had hilar lymphadenopathy. Signal loss was displayed on out-of-phase images compared to in-phase images in six (n_present_ = 6/10, 60%) patients. Significant differences in hepatomegaly and hilar lymphadenopathy were observed among the three imaging phenotypes in the present study group.

In reference group (Appendix A), only five cases with fibrotic signs were described, including patchy liver parenchyma changes (n_reference_ = 2) and the periportal halo sign (n_reference_ = 3).

## 4. Discussion

According to previous reports about imaging characteristics of hepatic LCH, the lesions are mainly divided into liver parenchyma and periportal abnormalities [4,26]. The majority of the descriptions of the imaging findings regarding histological stages depend on the CT density and MRI signal intensity (hypo- or hyperdense/signal intensity) [40]. Currently, imaging studies lack comparisons between adults and children, as well as a thorough investigation of imaging characteristics. The present study included both pediatric and adult patients with hepatic LCH. Initial CT/MRI scans were cross-sectionally analyzed, and follow-up examinations were longitudinally reviewed. In the present study, T1WI, T2WI, and DWI scans sufficiently illustrated liver lesions at various histological stages. Early proliferation presented as patchy and miliary lesions with blurred edges and slightly high signal intensities on T2WI/DWI. Granulomatous lesions tended to be cyst-like, rather than solid, on CT/MRI scans. The present study also focused on the evolution of lesions on MRI scans. Cyst-like lesions developed among patchy and miliary lesions; transformation from thick- to thin-walled cysts was associated with granulomatous regression; the regression of the periportal lesion is marked by decreased signal intensity and contracted periportal abnormality area. To sum up, the increase or decrease of lesions as well as the regression and transformation of lesions detected by MRI suggest that MRI can be applicable for the evaluation of treatment response and lesion activity during follow-up. Three manifestations of hypointense fibrosis on T2WI/DWI were summarized. In the present study, hepatic LCH was divided into three image phenotypes based on the distribution and morphology of the lesions seen on CT/MRI images. Differences in patient age, clinical features, and fibrosis were observed among the three types. We believe that this classification corresponds to the different clinical patterns of liver involvement. 

In addition to the analysis of cases from our institution, hepatic LCH cases from the relevant literature were also reviewed in the present study. No significant differences in basic characteristics were observed between the two groups (Table 1). In present study, there were more adult patients with liver involvement than pediatric patients. Since our institution is a large tertiary referral center in central China, pediatric cases in our institution are relatively incomplete, and the patient’s condition is more severe. The exact incidence of LCH in adults has not been reported [3]. Some studies believe that hepatic LCH in adults is very rare [45]. Perhaps due to a lack of awareness of liver involvement, the incidence of hepatic LCH is underestimated. Although hepatomegaly, liver biochemical abnormalities, and hepatic nodular abnormalities on images are recommended as criteria for liver involvement in guidelines derived from the Euro-Histio-Net project (2012 edition for pediatric patients and 2013 edition for adult patients) [6,7], hepatomegaly and liver biochemical abnormalities were absent in 37.3% and 26% of patients when combining both groups, respectively. For all patients whose initial medical information was available in the present study group, liver abnormalities were found on imaging scans at the initial visit. Li et al. [45] speculated that the liver may be involved earlier than other systems; therefore, improving the understanding of the imaging features of hepatic LCH is helpful in detecting it when clinical symptoms are absent.

### 4.1. Differences among the Three Imaging Phenotypes: Patient Age, Clinical Feature, and Fibrosis

Cases from both groups (n_present_ = 20, n_reference_ = 24, n_combined_ = 44) were classified into three types based on the distribution and morphology of the lesions on CT/MRI scans (Table 1 and Appendix A). Based on morphological changes and lesion distribution, the pathological findings of the three patients were consistent with the image phenotype. Due to the limitation of fine needle puncture, biopsy cannot assess whole liver as MRI can. The three image phenotypes revealed differences in patient age, clinical features, fibrosis, and other imaging findings (hepatomegaly and hilar lymphadenopathy), which helped radiologists and clinicians deepen their understanding of different patterns of liver involvement. Considering that patients may delay seeking medical care, we recommend using the imaging classification of patients as a reference for the initial liver assessment. 

Central periportal lesion phenotype

The central periportal lesion phenotype (n_present_ = 6, n_reference_ = 14, n_combined_ = 20) was characterized by periportal abnormal density/signal intensity without parenchymal lesions. This may reflect different types of infiltration patterns for histocytes or may be related to the small number of cases in this study. This phenotype was observed in young children in both groups. Biochemical abnormalities of this type were more apparent than those of the other two types in the present study. Initial imaging examination of cases of this phenotype revealed that the lesion was distributed along the portal vein. On follow-up images in the present study, all six patients (6/6, 100%) eventually developed hypointense areas on T2WI/DWI related to liver fibrosis, including the periportal halo sign (n_present_ = 2, 2/6, 33.3%), giant, central nodular fibrosis (n_present_ = 1, 1/6, 16.7%), and patchy parenchymal fibrosis (n_present_ = 3, 3/6, 50%). A previous clinicopathological study of 20 hepatic LCH patients described this phenotype as an expansion of the portal triads corresponding to variable degrees of portal triaditis, but the parenchyma was normal [46]. Similar findings have been described in the past based on early case reports [9,25,26,27,28,29,42,44]. 

Disseminated lesion phenotype

The lesions of disseminated lesion phenotype (n_present_ = 8, n_reference_ = 9, n_combined_ = 17) were diffusely distributed in the liver parenchyma and portal tracts, and periportal space widening was relatively mild. This phenotype was found in children, adolescents, and adults in both the groups. Lesions can change rapidly on follow-up images. Various histological phases can simultaneously exist. Accompanied by resolution of the lesion, patchy fibrotic hypointensity was observed (n_present_ = 3, 3/8, 37.5%). Heyn et al. [46] described this phenotype from a pathological view that eosinophilic granulomas are distributed throughout the liver parenchyma and portal tracts. Similar cases have been reported in the past, with relatively limited description of imaging findings, lack of long-term follow-up studies of patients, and description of fibrotic signs [4,13,14,15,16,17,18,24,26]. 

Scattered lesion phenotype

The scattered lesion phenotype (n_present_ = 6, 6/20, 30%) is characterized by several scattered lesions (generally < 15) in the liver parenchyma, which were found in adults. Hepatomegaly and biochemical liver abnormalities were mild or non-existent, and this phenotype developed slowly on follow-up imaging. No fibrotic signs were observed in this phenotype. Mampaey et al. [12] reported a case similar to present study. There are few reports of this phenotype because normal laboratory tests and asymptomatic tests make it easy to miss.

### 4.2. Evolution of Lesions

The interpretation of imaging data implicates four histological phases of hepatic LCH: proliferative, granulomatous, xanthomatous, and fibrous phase. 

The proliferative and granulomatous phases are characterized by histiocyte infiltration, inflammation, and edema, with lesions clustered in the portal tracts and lobules. A previous report described imaging findings at this stage as periportal hypodensity on CT, hypointensity on T1WI and hyperintensity on T2WI, and mild enhancement on enhanced sequences. The decrease in signal intensity on T2WI is related to transformation to the fibrous phase [10].

In the present study, in addition to periportal abnormal signal intensity (Figure 4), patchy, miliary, and cyst-like lesions of the liver parenchyma were observed (Figure 2, Figure 3, Appendix A and Appendix A). Proliferative lesions are patchy or miliary, with blurred edges and slightly high signal intensity on T2WI/DWI. 

As the lesions progress to the granulomatous stage, granulomas form. Cyst-like lesions on the images were confirmed to be granulomas with peripheral Langerhans cells and central eosinophils [26]. Follow-up MRI examinations in present study demonstrated the evolution of lesions: thick-wall cyst-like lesions developed from patchy and miliary lesions, accompanied by the patchy and miliary lesions regressing and disappearing gradually. Pathologically, cyst-like lesions were found on MRI scans corresponding to the stage when granulomatous lesions had formed. The regression of the thick-wall cyst-like lesions included their transformation into thin-walled cyst-like lesions and disappearance without passing through any intermediate stage. In a PET-CT study of liver involvement, Hu et al. [15] reported that thick-walled cysts on MRI scans revealed increased fluorodeoxyglucose (FDG) uptake associated with the hypercellularity of the granulomas, thin-wall cysts revealed reduced or absent FDG uptake on PET-CT associated with less cellularity and fibrosis, and follow-up MRI scans illustrated that thin-walled cysts increased and then regressed. Our finding that the evolution of granulomas is transformed from thick- to thin-walled is consistent with the findings of Hu et al. [15]. Morphology and signal changes revealed on MRI scans can evaluate the evolution of lesions as well as PET-CT.

### 4.3. Detailed Additions to Imaging Findings

The present study complemented the MRI findings of liver lesions. Shi et al. [26] reported the detection of cyst-like lesions and ring-like enhancements. Cases in the present study showed more details of cyst-like lesions: thick-wall cyst-like lesions at the early stage were demonstrated as thick and uneven walls with slightly high signal intensity on T2WI, miliary nodules with increased intensity were occasionally seen in the thick wall, and the intracyst appeared hyperintense on T2WI/DWI (Figure 2 and Appendix A). The transformation from thick- to thin-walled lesions occurred as the cyst wall gradually became thinner, and a annular low signal intensity on T2WI and DWI around the cyst-like lesion gradually appeared. On dynamic contrast-enhanced MRI, the cyst wall showed mild to moderate enhancement in the arterial phase, ring-like enhancement became clearer in the portal venous phase, persistent enhancement, and the area of enhancement increased in the delayed phase; hence, some tiny cysts appeared as nodular in the delayed phase.

Due to the lipid-rich nature of histocytes in the xanthomatous phase, liver lesions with focal fatty infiltration appear to have low attenuation on CT and to be hyperintense on T1WI and hypointense on T2WI [47]. Shi et al. [26] reported that the presence of heterogeneous signal intensity was due to the coexistence of inflammation, fat infiltration, and fibrosis. In the present study group, out-of-phase images showed inhomogeneous signal loss of periportal granular lesions and liver parenchymal lesions compared to in-phase images. We believe that the different histological stages may coexist. Xanthomatosis, however, tends to be present in old lesions. A comparison between the out-of-phase and in-phase images is more conducive for detecting focal fat infiltration.

### 4.4. Earlier Detection of Liver Fibrosis

It is considered a typical manifestation of late fibrosis with stenoses and dilatation of the bile ducts corresponding to sclerosing cholangitis on MRCP and the nodular appearance of liver cirrhosis [10,26,27,48]. Heyn et al. [46] reported that liver fibrosis occurs much earlier than cirrhosis.

Previous LCH case reports have described liver fibrosis as multiple low-intensity areas on T2WI [11,28]. Three morphological patterns (Figure 2 and Appendix A) of hypointense fibrosis on T2WI/DWI were observed in the present cases, including the periportal halo sign, central nodular fibrosis, and patchy fibrosis extending from the central to peripheral liver parenchyma. The periportal halo sign, common in primary biliary cirrhosis, was described as a low-signal intensity lesion on T1WI and T2WI centered around portal tracts, approximately 5 mm–1 cm in size [49]. Wenzel et al. [50] speculated that it contributes to periportal fibrosis or apoptosis. Giant nodular fibrosis in the central liver was similar to regenerative nodules, described as a regenerative pattern led by cholestasis [51]. Patchy fibrosis has been described in two previous case reports [11,28].

Considering the limitations of liver biopsy, using DWI and ADC measurements as part of a standard MRI protocol can provide additional information for evaluating patients with liver fibrosis [52]. Charatcharoenwitthaya et al. [38] utilized DWI to evaluate liver fibrosis in patients with chronic viral hepatitis. The final ADC of each patient with fibrotic hypointense changes on T2WI/DWI in the present study with a b-value of 1000 s/mm^2^ was <1.04 × 10^−3^ s/mm^2^, which was the optimal cutoff for significant fibrosis in a previous study using DWI to assess liver fibrosis in chronic viral hepatitis (METAVIR Fibrosis Stage ≥ 2) [38]. The final ADC value per patient without fibrotic hypointense change was greater than 1.04 × 10^−3^ s/mm^2^. The ADC value of the present cases was associated with fibrosis, but readers had a moderate agreement (ICC = 0.602), and this value corresponds closely to a prior study that used DWI to stage liberal fibrosis and assessed the interobserver variability [53]. ADC cannot be used solely for the diagnosis of liver fibrosis [54,55,56]. We suggest combining T2WI/DWI and ADC value measurements to comprehensively analyze fibrosis.

Early fibrosis may regress [57,58]; therefore, it is necessary to detect fibrosis at an early stage. The grading of the periportal halo sign was found to be significantly correlated with the histological stages of liver fibrosis in patients with primary biliary cirrhosis [59]. We longitudinally analyzed the previous imaging data of patients with typical fibrotic signs and found that localized, slightly low signal intensity on T2WI and DWI was observed around the portal vein or in the liver parenchyma before the formation of apparent fibrotic signs (Figure 4). Early manifestations are not obvious and can be easily ignored. We suggest that slight hypointensity on DWI/T2WI should be differentiated from the disappearance of local lesions.

### 4.5. Limitations

The present study, however, has some limitations. First, it was a retrospective study with a small group of patients due to the low incidence of this disease. Moreover, there may be a bias in selecting patients for CT and MRI. The guidelines, however, recommend CT/MRI for unclear sonographic pathology, highlighting one of the clinical situations in which CT/MRI is used for diagnosis. Additionally, the present study enrolled patients from our institution and from prior reports to provide a more comprehensive analysis. Second, owing to the retrospective nature of the present study, there were no uniform imaging modalities, examination sequences, or protocols for the patients. Currently, there is a lack of consensus regarding recommendations for imaging techniques to assess liver involvement. Previous reports have primarily described the imaging characteristics of CT, CECT, and MRI plain scans, as well as MRCP. The present study emphasized the role of MRI in characterizing hepatic LCH, and adds imaging descriptions from DWI and dynamically enhanced sequences. Third, the present study is a preliminary investigation of DWI combined with ADC in the evaluation of liver fibrosis in LCH, and ADC values for the prediction of liver fibrosis grade need further evaluation with a larger sample size.

## 5. Conclusions

Hepatic abnormalities of infiltrative lesion and liver fibrosis can be detected and well-characterized on routine abdominal MRI scans. Additionally, the evolution of lesions was well demonstrated on follow-up MRI scans. The detailed features added in the present study are helpful in increasing the awareness of hepatic LCH among radiologists and clinicians. CT does not provide as much detail as MRI in distinguishing lesions with various histological nature. We recommend routine abdominal MRI to detect liver involvement and assess disease activity and treatment response.

## Figures and Tables

**Figure 1 bioengineering-10-00598-f001:**
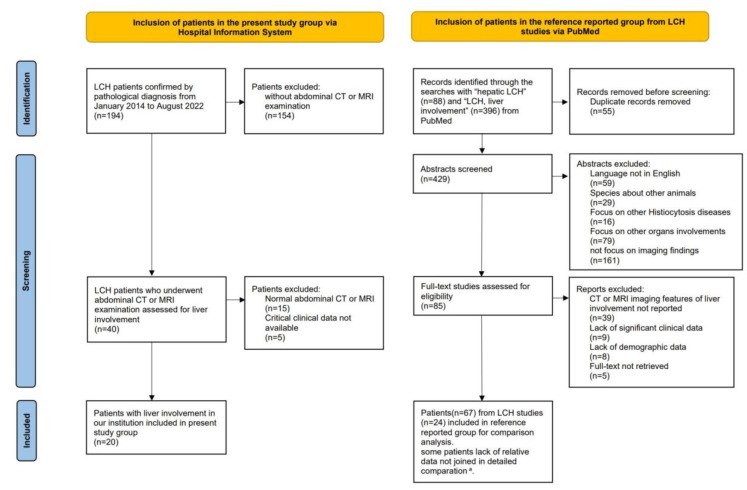
Flow-chart of the study population, following the PRISMA format. ^a^ Among the 67 patients in the reference group, data regarding hepatomegaly, liver biochemical abnormalities, and LCH stratification were available in 47 cases, 53 cases and 61 cases, respectively. The lack of relevant data was not included in the detailed comparisons.

**Figure 2 bioengineering-10-00598-f002:**
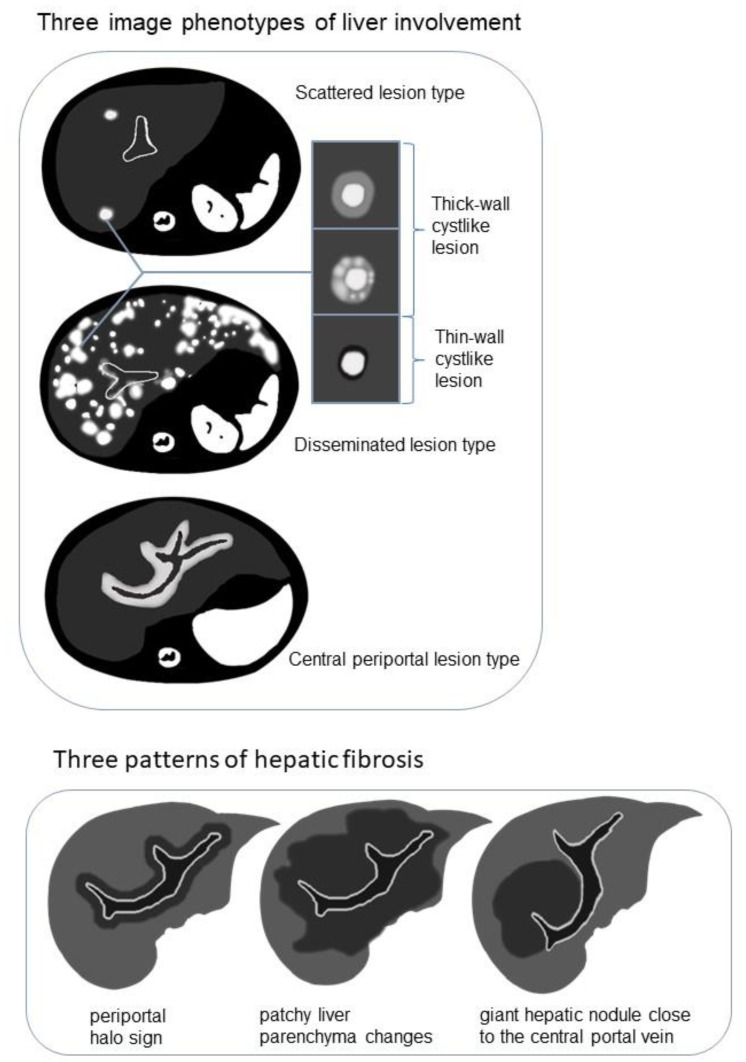
Schematic diagram demonstrating the image phenotypes of liver involvement with examples of cyst-like lesion and three patterns of hepatic fibrosis on T2-weighted imaging (T2WI)/diffusion-weighted imaging (DWI).

**Figure 3 bioengineering-10-00598-f003:**
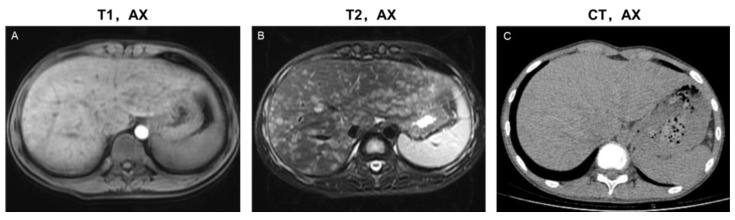
Imaging features of the disseminated lesion phenotype. A 14-year-old male teenager presented with intermittent abdominal pain for four months. The patient was diagnosed with single-system LCH (SS-LCH). Miliary and patchy lesions with cyst-like lesions among them distributed diffusely in the liver parenchyma and portal tract area. Based on this, he was classified as having a disseminated lesion phenotype. MRI images (**A**,**B**) revealed more details of the lesions than the CT images (**C**). T1, AX: Axial T1−weighted image; T2, AX: Axial T2−weighted image; CT, AX: Axial CT image.

**Figure 4 bioengineering-10-00598-f004:**
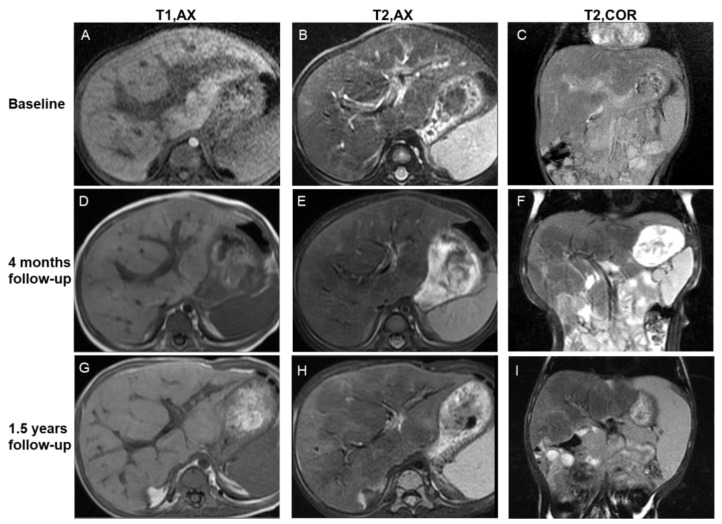
Imaging features of the central periportal lesion phenotype. A 12-month-old male infant presented with elevated liver enzyme levels. The patient was diagnosed with multisystem LCH (MS-LCH). MRI scans illustrate the evolution of the central periportal lesion phenotype that the periportal abnormal signal intensity gradually resolved, and patchy fibrotic hypointensity increased on T2WI. On initial MRI scans (**A**–**C**), slight periportal hypointense on T1WI and slight periportal hyperintense on T2WI corresponded to histiocytic infiltration and surrounding edema. Four months later (**D**–**F**), the wide periportal lesions disappeared, and the signal intensity decreased on T2WI. Patchy parenchymal hypointensity around periportal lesions related to fibrosis can be observed. Follow-up images after 1.5 years (**G**–**I**) show distorted portal tracts and distinct, markedly patchy hypointensity on T2WI. Coronal images (**C**,**F**,**I**) reveal the progression of signs of fibrosis more clearly. T1, AX: Axial T1−weighted image; T2, AX: Axial T2−weighted image; T2, COR: Coronal T2−weighted image.

**Table 1 bioengineering-10-00598-t001:** Basic characteristics of enrolled patients from both groups.

Characteristics	Combined ^a^	*p*-Value 1 ^b^	Reference Reported ^c^	Present Study	Scattered Lesion Type in Present Study	Disseminated Lesion Type in Present Study	Central Periportal Lesion Type in Present Study	*p*-Value 2 ^d^
Number of patients	87		67	20	6	8	6	
Sex, n (%)		0.851						1.000
Male	55 (63.2)		42 (62.7)	13 (65)	4 (66.7)	5 (62.5)	4 (66.7)	
Female	32 (36.8)		25 (37.3)	7 (35)	2 (33.3)	3 (37.5)	2 (33.3)	
Age at diagnosis, n (%)		0.620						<0.001 *
≥15 Y	52 (59.8)		41 (61.2)	11 (55)	6 (100)	5 (62.5)	0	
<15 Y	35 (40.2)		26 (38.8)	9 (45)	0	3 (37.5)	6 (100)	
Mean ± SD, (Years)	NA		NA	19.5 ± 16.5	36.8 ± 8.4	19.7 ± 12.9	1.8 ± 0.4	
Range, (Years)	NA		NA	1.1~53	29~53	1.1~39	1.2~2.5	
Liver involvement, n(%)		/						/
At baseline	NA		NA	18 (90)	4 (66.7)	8 (100)	6 (100)	
NA	NA		NA	2 (10)	2 (33.3)	0	0	
Hepatomegaly, n (%)	42 (62.7)	0.798	29 (61.7) ^e^	13 (65.0)	1 (16.7)	7 (87.5)	5 (83.3)	0.019 *
Liver biochemical abnormalities, n (%)	54 (74.0)	0.471	38 (71.7) ^f^	16 (80.0)	2 (33.3)	8 (100.0)	6 (100.0)	0.006 *
Stratification, n (%)		1.000						1.000
MS-LCH	78 (96.3)		59 (96.7) ^g^	19 (95)	6 (100)	7 (87.5)	6 (100)	
SS-LCH	3 (3.7)		2 (3.3)	1 (5)	0	1 (12.5)	0	
Other organ, n (%)		/						0.004 *
Bones	NA		NA	10 (50)	5 (83.3)	2 (25)	3 (50)	
Lung	NA		NA	8 (40)	2 (33.3)	5 (62.5)	1 (16.7)	
Skin	NA		NA	8 (40)	0	3 (37.5)	5 (83.3)	
Lymph node	NA		NA	4 (20)	1 (16.7)	0	3 (50)	
pituitary	NA		NA	6 (30)	1 (16.7)	5 (62.5)	0	
Thymus	NA		NA	1 (5)	0	1 (12.5)	0	
Thyroid	NA		NA	3 (15)	1 (16.7)	2 (25)	0	
bone marrow	NA		NA	2 (10)	0	1 (12.5)	1 (16.7)	
Mucosa	NA		NA	3 (15)	0	0	3 (50)	
Nervous system	NA		NA	2 (10)	1 (16.7)	1 (12.5)	0	
Ear	NA		NA	2 (10)	0	0	2 (33.3)	
Genitalia	NA		NA	1 (5)	0	1 (12.5)	0	
Spleen	NA		NA	3 (15)	0	1 (12.5)	2 (33.3)	

^a^ The sum of cases from the reference group and present study group for each item. ^b^ *p* value of comparison between the reference group and present study group. ^c^ A total of 24 articles with 67 patients enrolled in the reference group. ^d^ *p* value of comparison between three types of patients in the present study group. ^e^ Data of hepatomegaly were available in 47 cases from the reference group. Combined cases from two groups (n_combined_ = 67). ^f^ Data of liver biochemical abnormalities were available in 53 cases from the reference group. Combined cases from two groups (n_combined_ = 73). ^g^ Data of stratification were available in 61 cases from the reference group. Combined cases from two groups (n_combined_ = 81). “/”, defined as the relative comparison was not done. * A *p* value of <0.05 indicates a significant difference. MS-LCH: multisystem LCH; SS-LCH: single-system LCH; NA: not available.

**Table 2 bioengineering-10-00598-t002:** Liver biochemical results of enrolled patients from the present study group.

Liver Biochemical Results	Scattered Lesion Phenotype	Disseminated Lesion Phenotype	Central Periportal Lesion Phenotype	*p*-Value
Number of patients	6	8	6	
γGT, AP				<0.001 *
Normal	4	0	0	
γGT increased ≤2 times normal/AP increased ≤1.5 times normal	2	2	0	
γGT increased >2 times normal/AP increased >1.5 times normal	0	6	6	
ALT, AST				0.025 *
Normal	5	2	0	
ALT/AST increased ≤3 times normal	1	4	1	
3 times normal <ALT/AST increased ≤5 times normal	0	1	2	
ALT/AST increased >5 times normal	0	1	3	
Albumin				0.066
Normal	6	6	2	
Hypoalbuminemia >30 g/dl	0	2	4	
Hypoalbuminemia <30 g/dl	0	0	0	
Bilirubin				0.01 *
Normal	6	6	1	
Hyperbilirubinemia ≤3 times normal	0	1	4	
3 times normal < hyperbilirubinemia ≤5 times normal	0	1	0	
Hyperbilirubinemia >5 times normal	0	0	1	

Note: the liver biochemistry was graded based on the guidelines derived from the Euro-Histio-Net project [7]. * A *p* value of <0.05 indicates a significant difference.

**Table 3 bioengineering-10-00598-t003:** Other accompanied imaging signs of three imaging phenotypes.

Accompanied Imaging Signs	All Patients	Scattered Lesion Type	Disseminated Lesion Type	Central Periportal Lesion Type	*p* Value ^b^
Number of patients	20	6	8	6	
T2/DWI hypointense, n (%)					0.019 *
Periportal halo sign	2 (13)	0	0	2 (33.3)	
Giant, central nodular fibrosis	1 (6.7)	0	0	1 (16.7)	
Patchy fibrosis	6 (40)	0	3 (60)	3 (50)	
NA ^a^	5 (25)	2 (33.3)	3 (37.5)	0	
Hepatic surface, n (%)					0.035 *
Smooth	17 (85)	6 (100)	8 (100)	3 (50)	
Nodular	3 (15)	0	0	3 (50)	
Hepatomegaly, n (%)	13 (65)	1 (16.7)	7 (87.5)	5 (83.3)	0.019 *
Splenomegaly, n (%)	12 (60)	1 (16.7)	6 (75)	5 (83.3)	0.053
Enlarged lymph Nodes, n (%)	11 (55)	0	7 (87.5)	4 (66.7)	0.005 *

^a^ Five patients did not undergo MRI scan, so the T2/DWI signal intensity were not available, so they were not counted into percentages. ^b^ The accompanied imaging signs, including T2/DWI hypointense, hepatic surface, hepatomegaly, splenomegaly, and enlarged lymph nodes were compared among patients with three imaging phenotypes and relevant *p* values in the same row. * A *p* value of <0.05 indicates a significant difference. NA: not available.

## Data Availability

The datasets generated for this study can be found in the submitted article.

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
