# Peer review of "Imaging Phenotypes and Evolution of Hepatic Langerhans Cell Histiocytosis on CT/MRI: A Retrospective Study of Clinical Cases and Literature Review"

_bioengineering, 2023, doi:10.3390/bioengineering10050598_

Round 1
Reviewer 1 Report
The paper by Hao et al describes analysis of hepatic Langerhans cell histiocytosis on CT/MRI images. The theme of the study is important and interesting to a wide range of readers. The overall format and presentation of data is adeqate, and methods, results and references contains quite full description of required data. Thus, only some minor suggestions to improve the text:
1. Please, provide the name of your institution (eg "The present study retrospectively analyzed the clinical and imaging data of patients treated at this institution...") to give the readers the exact information.
2. In section 2.4 add references to exact statistical package that was utilized in the proposed calculations.
3. Figures 3 and 4 are missing dimentions. The same is true for images in supplementary file.
4. It seems like it is worth mentioning of exact values, obtained in this study, in the Conclusion section and in Results part of the abstract.
5. Please, check journal gudelines and correct minor discrepancies.
The paper may be published after correction of mentioned issues.
Reviewer 2 Report
This manuscript reports the results of a retrospective study into the use of computed tomography (CT) and magnetic resonance imaging (MRI) to image the progression of hepatic Langerhans cell histiocytosis across a sample set of patients from both the authors’ own institution and from previously published papers on the topic. The manuscript is generally well-written and the comparative analyses conducted are reasonable and thorough. Whilst the study does suffer from a limited sample size, this is likely an inevitability given the low incidence of the disease. It is likely that the conclusions drawn and insights provided would be of use to medical imaging scientists. There are some points for the authors to address and these are detailed below.
1. Table 3. The positioning of some of the p-values in the table are such that it makes it hard for the reader to quickly see which comparisons between values (i.e. which null hypotheses) the p-values apply to. Can you clarify the presentation to make these connections clearer?
2. Page 9 lines 226-228 “. . . had a wider zone . . . (1.8 + 0.4 Y).” Are the errors presented here standard deviations or standard errors? It would be helpful to mention this somewhere in the manuscript.
3. Page 12 line 346 “. . . the signal intensity . . . of regression too.” This passage is unclear and needs to be reworded.
4. Page 12 lines 354-355 “Adult patients more . . . than pediatric patients.” Do you have any insights into why this is the case? Can you comment in the manuscript on possible reasons?
5. Page 13 lines 427-428 “. . . included direct disappearance . . . cystic-like lesions.” What do you mean by “direct disappearance” in this context? Do you mean immediate disappearance or disappearance without passing though any intermediate stage/s? Clarify your meaning in the manuscript.
6. Page 15 lines 506-507 “. . . imaging sequences to assess liver involvement.” By “imaging sequences” do you mean which imaging techniques are used when or how frequently the images are taken or how many images are taken? It would be helpful to be more specific here.
Generally the manuscript is well-written but there are certain passages where the syntax is confusing and unclear. Also, the authors repeatedly use the term "on behalf of" when they likely mean "caused by".
Reviewer 3 Report
There are three minor comments.
It would be better to explain the relationship between the three image types and pathological findings.
It would be better to explain why cyst-like lesions are found.
It would be better to add whether there is a difference between early and late lesions in CT and MRI images, respectively.
Please check English grammar.
